# Visualization of π-hole in molecules by means of Kelvin probe force microscopy

B. Mallada [1,2,3,6], M. Ondráček [1,6], M. Lamanec [3,4,5,6], A. Gallardo[1],
A. Jiménez-Martín[1,2], B. de la Torre [1,2] ✉, P. Hobza [4,5] ✉ & P. Jelínek [1,2] ✉

Submolecular charge distribution significantly affects the physical-chemical properties of molecules and their mutual interaction. One example is the presence of a π-electron-deficient cavity in halogen-substituted polyaromatic hydrocarbon compounds, the so-called π-holes, the existence of which was predicted theoretically, but the direct experimental observation is still missing. Here we present the resolution of the π-hole on a single molecule using the Kelvin probe force microscopy, which supports the theoretical prediction of its existence. In addition, experimental measurements supported by theoretical calculations show the importance of π-holes in the process of adsorption of molecules on solid-state surfaces. This study expands our understanding of the π-hole systems and, at the same time, opens up possibilities for studying the influence of submolecular charge distribution on the chemical properties of molecules and their mutual interaction.

Non-covalent interactions are crucial in many chemical and biological processes, such as supramolecular assembling, ion recognition, and protein stability. The nature of non-covalent intermolecular bonds is determined, among others, by dispersion and electrostatic forces. While the dispersion force is always attractive and directionless, the electrostatic component can be attractive or repulsive and is highly directional. The nature of the electrostatic interaction is intimately linked to the internal charge distribution within the molecule. Therefore, precise knowledge of the charge distribution in molecules is a fundamental requirement for comprehending non-covalent interactions[1].

For instance, in the most prevalent category of molecular species, polycyclic aromatic hydrocarbons (PAHs), the larger electronegativity of carbon compared to the peripheral hydrogen results in an accumulation of electron density in the delocalized/conjugated π-bonds on the carbon skeleton. This electron-rich delocalized/conjugated π-bond system, which is evenly distributed above and below the molecular plane, generates a negative quadrupole moment for the PAH

molecules, see Fig. 1. However, substituting peripheral hydrogens with other substituents that are more electronegative than carbon, such as fluorine or chlorine, reverses the electron population of the π-bond system to make it electron-deficient[2], see Fig. 1. Consequently, the quadrupole moment of such a molecule becomes positive[3]. The presence of a π-electron-deficient cavity in the central part of the molecule is called a π-hole[4,5].

The π-hole concept can be vividly illustrated by comparing the electron distribution of benzene, $C_6H_6$, and hexafluorobenzene, $C_6F_6$. In benzene, electrons are localized within the aromatic carbon skeleton, as depicted in the electrostatic potential map shown in Fig. 1. Conversely, in hexafluorobenzene, the higher electronegativity of fluorine results in electron withdrawal from carbon atoms towards halogens, leading to a depletion of electron density in the central π-system on the carbon atoms. The origin of the distinct localization of the electron density in both systems is closely associated with the different characteristics of occupied molecular orbitals. Namely, in the case of $C_6F_6$, the occupied orbitals are delocalized over both carbon

[1]Institute of Physics, Academy of Sciences of the Czech Republic, Prague, Czech Republic. [2]Regional Centre of Advanced Technologies and Materials, Czech Advanced Technology and Research Institute (CATRIN), Palacký University Olomouc, 78371 Olomouc, Czech Republic. [3]Department of Physical Chemistry, Palacký University Olomouc, Tr. 17. listopadu 12, 771 46 Olomouc, Czech Republic. [4]Institute of Organic Chemistry and Biochemistry, Czech Academy of Sciences, Flemingovo Náměstí 542/2, 16000 Prague, Czech Republic. [5]IT4Innovations, VŠB – Technical University of Ostrava, 17. Listopadu 2172/15, 708 00 Ostrava-Poruba, Czech Republic. [6]These authors contributed equally: B. Mallada, M. Ondráček, M. Lamanec ✉e-mail: bruno.de@upol.cz; pavel.hobza@uochb.cas.cz; jelinekp@fzu.cz

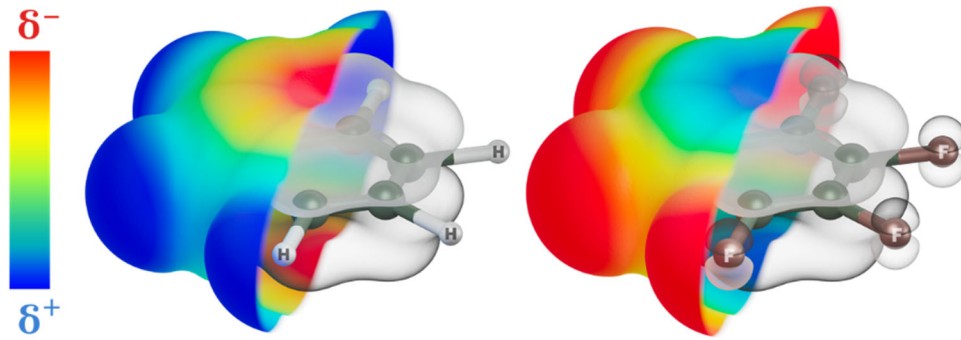

**Fig. 1 | Schematic description of the concept of π-hole.** The spatial distribution of the highest occupied molecular orbitals (HOMO) (grey-shaded volumes) of benzene ($C_6H_6$) on the left and hexafluorobenzene ($C_6F_6$) on the right overlapped with the electrostatic potential map (coloured surface from red (negative charge) to blue (positive charge)).

and fluorine atoms. In contrast, in the case of $C_6H_6$ molecule, they are predominantly localized on carbon atoms only, as seen in the grey surfaces in Fig. 1.

As discussed above, the presence of a positively charged π-hole significantly affects non-covalent intermolecular interactions. For example, intermolecular stacking interactions[6], which play an essential role in the stabilization of the helical structure of DNA bases, can be strongly affected by the presence of a π-hole. The decisive role of the electrostatic interaction on the stacked structure can be well documented for the simplest stacked aromatic systems, dimers of benzene $C_6H_6$ and/or hexafluorobenzene, $C_6F_6$. In the case of homodimers, their stabilization is entirely caused by dispersion interaction, as the electrostatic quadrupole-quadrupole interaction is repulsive. However, in the case of heterodimers, the situation changes due to the attractive electrostatic quadrupole-quadrupole interaction, which allows a close approach of the aromatic rings resulting in significant dispersion interaction and, consequently, resulting in substantial stabilization of the heterodimers[3].

It is evident that the presence of π-hole may strongly affect the physicochemical properties of molecular systems. So far, the concept of π-hole has been developed exclusively on a theoretical basis using quantum calculations. Experimentally, the existence of π-hole has only been proven indirectly based on measured data, the interpretation of which can only explain the presence of π-hole[7] However, a direct experimental observation that would clearly prove the existence of the π-hole has been lacking so far.

Of all the experimental techniques, scanning probe microscopy (SPM) emerges as the most appropriate tool for the direct visualization of π-holes in molecular systems. In recent years, the development of SPM with a functionalized probe[8,9] has enabled the unprecedented spatial resolution not only of the chemical[8] and spin[10] structure of molecules on the surfaces but also of the anisotropic atomic charge on individual atoms, the so-called σ-hole[11]. In this article, we will show that the Kelvin probe force microscopy (KPFM) method, which enabled us to resolve the σ-hole for the first time, is an ideal imaging tool for the real space observation of the π-hole system as well.

## Results and discussion
### Experimental results
Here, we experimentally visualize the π-hole in 9,10-Dichloroocta-fluoroanthracene $C_{14}F_8Cl_2$ (FCl-An) molecule and, at the same time, its absence in anthracene $C_{14}H_{10}$ (An). As discussed above, the presence of different substituents rules the charge transfer between π-orbitals of the carbon skeleton and the substituents. Supplementary Fig. 1 displays selected occupied orbitals of An and FCl-An, whose comparison reveals their different shapes. Similarly, as in the case of $C_6H_6$ and $C_6F_6$, the molecular orbitals in An are localized only at the carbon skeleton, while these in FCl-An are also extended at all halogen

substituents. In the latter case, this gives rise to the presence of π-hole localized on the carbon skeleton. The presence of π-hole could also be seen in electron density difference maps[12], see Supplementary Fig. 2.

Here we investigated heterogeneous molecular self-assemblies consisting of FCl-An and An molecules. Figure 2a displays the molecular self-assembly grown by simultaneous sublimation of the two compounds on an atomically clean Au (111) surface kept at room temperature in an ultrahigh vacuum. Scanning probe microscopy images, acquired at a base temperature of 4.8 K, reveal large-scale ordered molecular islands composed of both molecular species alternating in a periodic fashion (Fig. 2a, b).

To determine the detailed arrangement of the molecular species in the assembly, we use noncontact atomic force microscopy (nc-AFM) with a functionalized carbon monoxide (CO) tip. The high-resolution nc-AFM images acquired at a constant height mode in Fig. 2b clearly distinguish two different molecular species. To gain insight into the self-assembly structure, we carried out total energy density functional theory (DFT) calculations as well as AFM imaging of the supramolecular structure on the Au (111) surface. The perfect agreement between experimental and theoretical nc-AFM images allowed us to unambiguously resolve the chemical structure of both An and FCl-An molecules, as well as their arrangement. The supramolecular assembly comprises a rhombic periodic structure with one An and two FCl-An molecules in the unit cell, as shown in Fig. 2b,c. The arrangement is dictated by attractive intermolecular electrostatic interactions between negatively charged fluorine atoms, σ-hole on chlorine, and positively charged hydrogen atoms (Fig. 2d). This scenario is supported by the fact that the mere deposition of An on the surface does not result in the formation of self-organized molecular structures, as shown in Supplementary Fig. 3.

Notably, the high-resolution AFM images show, apart from the difference in the apparent size of the molecular species, a relatively similar contrast of the carbon skeleton for both types of molecules. In addition, AFM images show that FCl-An molecules are systematically imaged brighter than An molecules due to a topographic effect, as we will discuss below. Thus, the AFM images apparently do not provide any direct experimental evidence of π-hole.

Therefore, we employed Kelvin Probe Force Microscopy, which records the spatial variation of the local contact potential difference (LCPD) across the surface[13]. At short tip-sample distances, the magnitude of the LCPD is affected by a short-range electrostatic interaction between the tip apex and surfaces, enabling us to map the charge distribution (See Fig. 3a, d) with atomic resolution[11,14]. The KPFM technique was employed to investigate the charge distribution at the atomic and molecular levels. This technique is useful for the detection of single electron charge states of individual atoms[15] and molecule[16], mapping charge distribution within molecules[17], resolving molecular dipolar moments[18], and visualizing the anisotropic charge

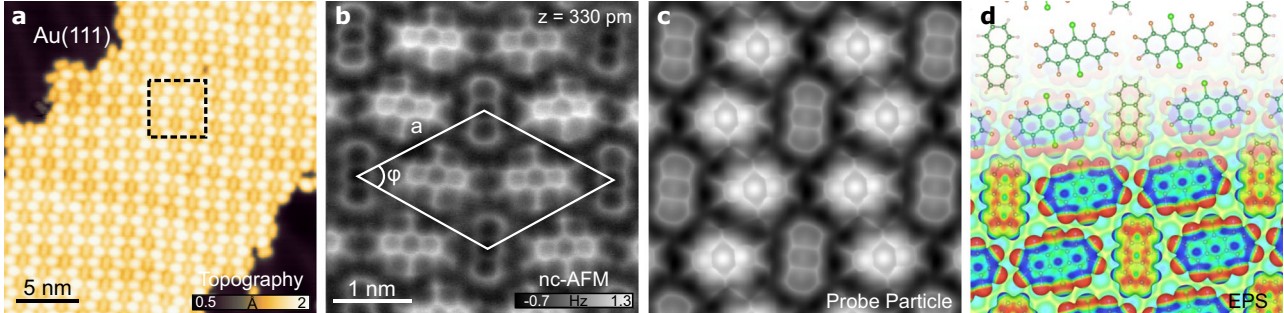

**Fig. 2 | Overview and description of the system. a** STM ($V_{Bias}$ = 500 mV, $I_{tunnel}$ = 10 pA) topography overview of the ordered self-assembly structure of An and FCl-An molecules on Au(111). **b** Constant height nc-AFM with CO-tip of the unit cell of the self-assembly (white solid line, $a$ = 1.85 ± 0.02 nm, $\varphi$ = 122 ± 0.6°). **c** Probe Particle simulated nc-AFM image. **d** Model of the self-assembly of An and FCl-An (top) and the electrostatic potential map of the optimized supramolecular structure (bottom). Source data are provided as a Source Data file.

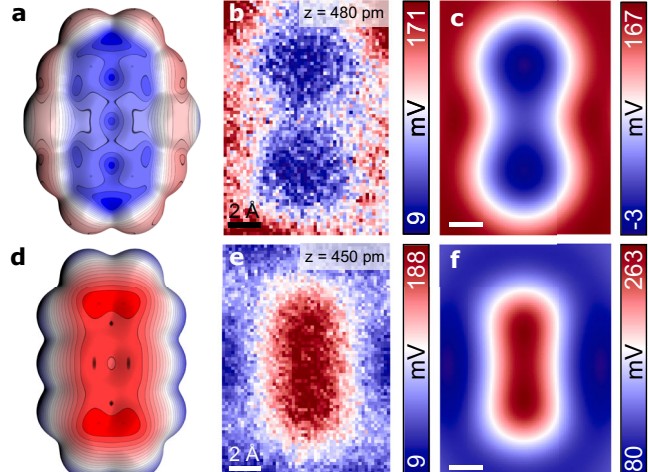

**Fig. 3 | Simulated and experimental charge distributions of FCl-An and An. a, d** Electrostatic potential maps of free-standing FCl-An and An molecules. **b, e** Experimental LCPD maps of FCl-An and An acquired with the same CO-tip in constant height mode. **c, f** Probe Particle simulated LCPD maps of FCl-An and An. Source data are provided as a Source Data file.

distributions in single atoms[11]. In previous work, it was demonstrated that the sensitivity of the KPFM method could be substantially enhanced[11,17,19,20] using functionalized probes. Moreover, it was demonstrated that the KPFM technique could capture contrast inversion for fluorinated and hydrogenated benzene rings within the same molecules[21]. These results indicate the possibility of resolving π-hole by the KPFM technique.

Figures 3b and 3e depict constant height KPFM maps conducted with the same CO-terminated tip of individual An and FCl-An molecules, respectively. These maps reveal a striking difference in the LCPD contrast between the two molecular species. Specifically, the LCPD signal over An/FCl-An molecule is shifted to a higher/lower value, indicating a decrease/increase in the local work function. The anthracene molecule possesses a uniform positive LCPD signal, representing a negative charge distribution. At the same time, the fluorinated counterpart exhibits a bow-tie shape of negative LCPD, indicating a positive charge distribution. Large KPFM images (Supplementary Fig. 4) of molecules adsorbed at various sites demonstrate the same trend, suggesting that the observed effects are not attributable to spatial variations in the LCPD of the substrate.

In terms of KPFM measurements, it is important to note that the obtained contrast is highly dependent on the tip-sample distance, as illustrated in Supplementary Fig. 5. Specifically when the tip-sample

distance is far, the KPFM images do not exhibit intramolecular features. As the tip-sample distance decreases, submolecular contrast becomes increasingly pronounced. Nonetheless, tip relaxations can noticeably influence the KPFM signal at shorter tip-sample distances and generate measurement artifacts. In contrast to nc-AFM imaging, where tip-relaxations improve the submolecular resolution, in KPFM measurement, such relaxations cause a noticeable deviation in the LCPD signal from its parabolic shape[22]. Therefore, we opt for a tip-sample height close to the Δf minima on both molecules, which ensures the absence of tip relaxation and enhances the contrast. This choice is further supported by the absence of submolecular resolution in Δf* images (Δf* corresponds to the Δf(V) maximum, that is, Δf at compensated LCPD, see Methods) for either An or FCl-An, as shown in Supplementary Fig. 5 and 6.

To rationalize the experimental KPFM images, we performed KPFM simulations using the optimized supramolecular structures obtained from DFT calculations using PP-AFM code[11,23]. The simulated KPFM images shown in Fig. 3c,f match very well with the experimental evidence. In particular, they also reproduce submolecular variation of the LCPD signal with the characteristic bow-tie pattern presented in the central part of FCl-An molecule. This internal variation of the LCPD signal also nicely matches the distribution of the electrostatic potential calculated for a free-standing FCl-An molecule, shown in Fig. 3a. This effect is associated with the heterogeneous charge distribution of the π-hole due to the different electronegativity of fluorine and chlorine modulating the charge transfer from the π-system locally.

It should be noted that KPFM measurements on the atomic scale are only qualitative and themselves cannot say anything about the sign of the charge on the substrate. To prove that the observed variation of LCPD contrast in the central part of the FCl-An molecule corresponds to the presence of the positive charge associated with π-hole, we take advantage of the presence of a positively charged σ-hole on Cl, see Fig. 2d. The σ-hole can be partially visualized when the probe is sufficiently close to the substrate, see Supplementary Fig. 7. Importantly, both the central region of the molecule and σ-hole show very similar LCPD vales -110 mV. This direct comparison enables us to confirm the presence of a positively charged region associated with π-hole. Optionally, the presence of a positive charge in the central part of the FCl-An molecule is also indirectly supported by the good agreement between the range of LCPD values in the experimental and simulated patterns shown in Fig. 3b, c.

### The influence of π -hole in the adsorption height

We already mentioned that the presence of a π-hole can substantially change the stacking interaction. Thus, we analysed how the presence of π-hole affects the adsorption of the molecules on a metallic substrate. A series of constant height AFM images, shown in Supplementary Fig. 8, reveal that submolecular contrast first emerges on FCl-An

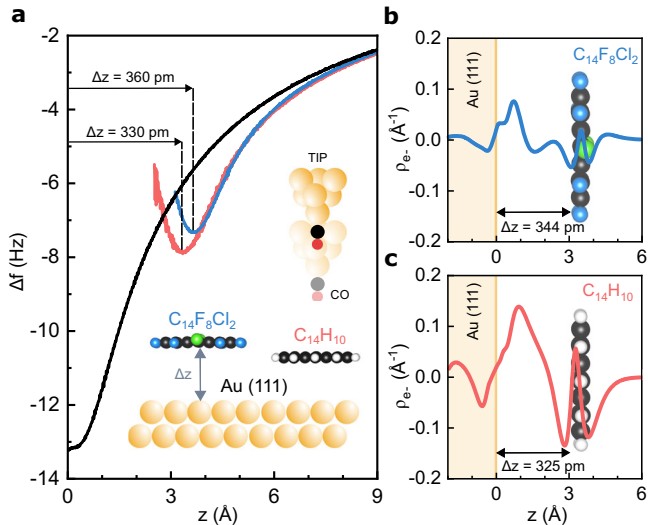

**Fig. 4 | Charge transfer and π-hole effect on the adsorption height.**
**a** Experimental adsorption heights estimated with $\Delta f$-z spectroscopies in the middle hexagon of the molecules (inset) FCl-An (blue line) and An (red line) respect to Au(111) substrate (black-line), see Supplementary Fig. 9. **b, c** Calculated induced electron density of FCl-An (top, blue line) on Au(111) and An (bottom, red line) as a function of the distance between the surface and the molecule. Source data are provided as a Source Data file.

molecules independently of their adsorption site. This observation suggests different adsorption heights of the FCl-An and An molecules on the Au(111) surface. To analyse in more detail the different adsorption heights of the molecules, we carried out specific-site force-distance spectroscopy. Figure 4a displays $\Delta f$-z spectroscopy acquired with the same CO-tip above the central benzene unit of both An and FCl-An molecules. The difference between the minima of two $\Delta f$-z curves of 30 pm can be directly related to the difference in the adsorption height of the molecule[24]. Moreover, we also acquired $\Delta f$-z spectroscopy over the bare Au(111) surface, which enables us to determine the relative height of the molecules above the Au(111) surface, to be 330 and 360 pm for An and FCl-An, respectively. These values match reasonably well with the adsorption height obtained from the total energy DFT calculations of molecular assembly on Au(111) surface (325 and 344 pm, respectively).

One may argue that the greater adsorption height of FCl-An molecule is caused by the presence of chlorine atoms in FCl-An. A larger atomic radius of chlorine atoms (1.75 Å (Cl) and 1.47 Å (F)) may enhance Pauli repulsion pushing the molecule out of the Au(111) surface. However, the total energy DFT simulation of the fully fluorinated molecule provides a remarkably similar adsorption height of 343 pm over the Au(111) surface, see Supplementary Table 1 Thus, the presence of chlorine atoms does not influence the adsorption height. Next, we looked at the character of bonding interaction between the molecules and metallic substrate. According to DFT calculations, both molecules are physiosorbed with negligible hybridization of molecular orbitals (see Supplementary Figs. 10 and S11), and the interaction is dominated by dispersion interaction, see Supplementary Table 2. Figure 4b, c represents the calculated induced electron density between An and FCl-An molecules and the Au (111) surfaces. We observe that there is more significant induced electron density in the interface between An molecule and metallic substrate compared to FCl-An/Au(111) interface. The induced charge density is a consequence of attractive electrostatic interaction between the molecule and the surface. In the case of An molecule, the HOMO orbital is found close to the Fermi level of metal, which facilitates the charge induction upon the adsorption (Supplementary Fig. 10). However, in the case of the FCl-An molecule, the presence of π-hole increases the ionization potential and consequently

suppresses the charge density induction at the molecule/metal interface.

To get more detailed insight into the adsorption energy, we perform the symmetry-adapted perturbation treatment (SAPT0), which provides the interaction energy decomposition scheme[25]. The SAPT0 analysis was performed on a cluster model, which gives very similar results to the slab model. It is also worth noting that DFT and SAPT0 calculations give very similar results of adsorption energies of molecules on Au(111), see Supplementary Table 2. Supplementary Table 3 presents the SAPT0 energy decomposition for all three molecules. It is evident that in all cases, the attractive interaction dominates the dispersion energy compared to the electrostatic and induction terms. Importantly, the induction energy describing stabilization energy due to electrostatic interaction between permanent charge multipoles and the induced charge is larger for An than for FCl-An molecule. This trend fully agrees with the above-mentioned findings on the relative magnitude of induced electron densities in An and FCl-An predicted by the DFT slab calculation, see Fig. 4b,c.

Here, we presented the experimental imaging of π-hole in molecular systems by its real space imaging on a single molecule by means of Kelvin probe force microscopy. We showed that the presence of π-hole also influences the electrostatic interaction between the molecule and a metallic surface and, consequently, the adsorption height of the molecule. Namely, the π-hole increases the ionization potential of the molecule, which inhibits the induction of the charge density at the molecule/metal interface. This results in weakened induced electrostatic interactions and, consequently, in larger adsorption height of the molecule. This study shows the potential of scanning microscopy for studying the internal charge distribution in molecules, which fundamentally affects their physical and chemical properties.

## Methods
### Experimental methods
The experiments were conducted at a temperature of 4.2 K using a commercial STM/nc-AFM microscope (Createc GmbH). Pt/Ir tips, sharpened by focused ion beam (FIB), were utilized and were further cleaned and shaped by gentle indentation (-1 nm) in the bare metallic substrate. STM topography was acquired in constant current mode with the bias voltage applied to the sample. In nc-AFM imaging, a qPlus sensor (resonant frequency ≈ 30 kHz; stiffness ≈ 1800 N/m) was operated in frequency modulation mode with an oscillation amplitude of 200 pm. Both nc-AFM and KPFM images were captured in constant height mode. The Au(111) substrate was prepared by repeated cycles of Ar+ sputtering (1 keV) and subsequent annealing at ~800 K. The STM/nc-AFM/KPFM images were processed using WSxM software[26].

### Molecular deposition
9,10-Dichlorooctafluoroanthracene (FCl-An) and anthracene (An) molecules were sublimated with two Knudsen cells simultaneously in UHV conditions at RT on the sample of Au (111) for 15 s.

### KPFM characterization and analysis
LCPD maps were obtained by fitting a parabolic expression to frequency shift vs bias spectroscopies ($\Delta f$(V, x, y)) collected at all the points of a 64 × 80 pixels rectangular grid of size 3 nm × 5 nm. The data was collected at constant height using the same CO functionalized tip for both anthracene and chlorinated molecules. Each data point was acquired in approximately 3 s, with each parabola containing 600 points. Following the acquisition, we fitted the data with a parabolic expression of the form $\Delta f(V) = a \times (V - LCPD)^2 + \Delta f^*$. The LCPD value, which corresponds to the bias at which the parabola has a maximum, and the $\Delta f^*$ parameter, which is the maximum frequency shift value, were extracted from the fitted parabolas and plotted separately in the (x, y) grids to generate maps. The data did not exhibit any significant distortion or deviation from expected parabolic behaviour within the

bias range of (−150, 400) mV for either molecule or tip. The acquisition height was chosen to be in the attractive regime to maximize the electrostatic contribution for both molecules.

## Theoretical methods

The optimized geometries of free-standing molecules, individual molecules adsorbed on the Au(111) surface, and the supramolecular assembly, both free standing and on-surface, were calculated using DFT-based methods implemented in the FHI-AIMS (Fritz Haber Institute Ab-Initio Materials Simulation) code[27]. Local atom-centred basis sets consisted of the "light" default basis sets (the version from 2010) for individual species as distributed with FHI-AIMS. The PBE[28] version of the GGA was employed for the exchange-correlation function. The surface Brillouin zone was represented by only one k-point (Γ). This lightweight setup – "light" basis, PBE functional, no k-grid – was needed to cope with the relatively large supercells in calculations that involved the Au(111) surface. For individual free-standing molecules, calculations with the hybrid PBE0[29,30] functional and the "tight" basis have also been carried out without substantial changes in either the geometry or charge distribution. The Au(111) slab consisted of 4 atomic layers. The size of the surface supercell intended to represent isolated molecules on the surface corresponded to the 6 × 6-unit cell of the unreconstructed Au(111) surface. In the case of the supramolecular structure, we first optimized the size of the supercell without the Au surface. Then, we constructed a √39 × √39 surface cell of Au(111) and deformed it somewhat (stretched by 1.20 % in one main crystallographic direction, compressed by 1.06% in the other, and reduced the angle between the two directions from 60° to 56.3°) so as to exactly match 2 optimal unit cells of the supramolecular structure. During geometry optimization, the free-standing molecules were forced to remain planar while for molecules on the surface, all atoms in the system except the bottom layer Au atoms were allowed to relax. Van der Waals correction of interatomic forces based on the Tkatchenko-Scheffler method with Hirshfeld partitioning[31] was applied to the molecular structures on the surface, excluding direct Au-Au interaction. The geometries were considered fully optimized when all forces except those corresponding to a constraint were under 2 meV/Å.

The AFM and KPFM images have been simulated using the Probe Particle (PP) model[23] adapted to a CO-terminated tip. The elastic deflection of the CO molecule has been modelled using the lateral (along the surface) spring constant of 0.25 N/m. DFT-calculated electron densities of the sample surface have been used in the PP model to evaluate the Pauli repulsion[32]. Furthermore, DFT-calculated maps of electrostatic potential and electrical polarizability (derived from the difference of electron densities in the external field of 0.1 eV / Å and without the external field, respectively) have been employed to generate the simulated KPFM maps[11] while assuming the $d_z^2$-like charge quadrupole of $-0.2$ e / Å$^2$ for the CO tip termination. The long-range contribution of the LCPD signal was adopted from the experimental $\Delta f(V)$ taken in the far tip-sample distance.

The molecular orbitals, as well as the electrostatic potential maps of isolated An and FCl-An were calculated for MP2/cc-pVTZ[33] optimized geometries. The cluster calculations used for SAPT0 analysis consisted of a single molecule An and FCl-An and surface represented by one layer made of 38 gold atoms with (111) orientation. Both studied molecules were optimized on the gold layer using PBE0[29] functional with Grimme's D3[34] dispersion correction using the Becke-Johnoson damping[35] and def2-TZVPP[36] basis set. The molecule was fully relaxed while the atoms of the gold layer were fixed. All these calculations were carried out by ORCA quantum chemistry program package[37]. SAPT0 calculations were made on the cluster models by PSI4[38] program in cc-pVDZ (cc-pwCVDZ-PP[39] for Au) basis set.

## Data availability

Source data are provided with this paper. The data supporting this study's findings are also available from the authors on request and in the repository https://doi.org/10.6084/m9.figshare.23660622. Source data are provided with this paper.

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

## Acknowledgements

We acknowledge the financial support of Czech Science Foundation GACR 20-13692X (A.G., B.M., M.O., P.J.), 19-27454X (P.H.), 23-06781 M (B.T.) and CzechNanoLab Research Infrastructure supported by MEYS CR (LM2023051). B.M. and M.L. acknowledge support from the Internal Student Grant Agency of the Palacký University in Olomouc, Czech Republic, IGA_PrF_2022_026 (B.M.) and IGA_PrF_2023_018 (M.L.). B.M. also acknowledges the Fischer scholarship. The authors gratefully acknowledge the support of the Operational Programme for Research, Development, and Education of the European Regional Development Fund (Project No. CZ.02.1.01/0.0/0.0/16_019/0000754). M.O., A.G and P.J. acknowledge computational resources provided by the e-INFRA CZ project (ID:90254), supported by the Ministry of Education, Youth and Sports of the Czech Republic.

## Author contributions

Conceptualization: P.J., P.H. Methodology: B.M., B.T., P.J. Theoretical calculations: M.O., A.G., M.L., P.J., P.H. Experimental: B.M., A.J-M., B.T. Funding acquisition: P.H., P.J., B.T. Supervision: B.T., P.H., P.J. Writing – original draft: B.M., B.T., P.H., P.J.

## Competing interests

The authors declare no competing interests.
