## [Peer Review File · Nature Communications]

Visualization of π -hole in molecules by means of Kelvin probe force microscopyREVIEWER COMMENTS

Reviewer #1 (Remarks to the Author):

This article describes the experimental observation of the pi-hole, and pi-hole is indeed an important concept, whether in guiding noncovalent interactions or in solid surface adsorption, it has also shown its importance in the field of crystal engineering. The experiments in the article are completed, and the discussion on results is also sufficient.

However I noticed that the experimental results of the article (Fig3) actually correspond to the electron density. Therefore, due to the lack of direct correspondence between the change of electrostatic potential map of molecule and the change of electron density (J. Chem. Theory Comput. 2009, 5, 2301 – 2312), I suggest that all the molecular surface electrostatic potential maps be changed to "electron density difference maps", or to present the electronic density difference maps together with the molecular surface electrostatic potential map, which is not easy to cause confusion for readers.

Another issue is that when the hydrogen atom is replaced by the fluorine atom, it is not necessary to emphasize the decrease of pi-electron density, but the decrease of electron density along the central position perpendicular to hexafluorobenzene pi-plane.

Finally, the article in communication form can be accepted for publication after minor revision.

Reviewer #2 (Remarks to the Author):

The manuscript reports the direct visualization of pi-electron-deficient cavity (referred in the following as a pi-hole) in a 9,10-Dichlorooctafluoroanthracene (FCI-An) molecule deposited on a Au(111) surface using atomic force microscopy (AFM). This work seems to be the first direct experimental observation of the existence of a pi-hole, and therefore worth publishing in a high-profile journal. The observation is made by a carefully planned set of experiments combining scanning tunneling microscopy, and probe-functionalized non-contact atomic force microscopy operated in constant-height imaging and Kelvin probe force modes. The experimental results seem to be fairly supported by first principles calculations. To demonstrate that the observation of the pi-hole at the FCI-An molecules is not due to the interaction with the substrate, anthracene (An) molecules are co-adsorbed on the surface and the AFM signals are simultaneously compared over the self-assembled structure formed by the two molecules.

Both experimental and theoretical results are convincing and support, in my view, the conclusions raised by the authors. I therefore recommend publication of the manuscript in Nature Communications upon minor revisions.

Fig.S2 does not fully support that the mere deposition of An molecules does not result on the formation of self-organized molecular structures, as the coverage of An molecules in Fig.S2 seems to be at least 50 times lower than the coverage of An molecules in Fig.1. I wonder whether structures of An molecules could be formed when the molecules are deposited on a gold surface at room temperature at similar coverage to the one in Fig.1.

It is stated that "The anthracene molecule possesses a uniform positive LCPD signal, representing a negative charge distribution, while the fluorinated counterpart exhibits a bow-tie shape of negative LCPD, indicating a positive charge distribution". However, the experimental LCPD value seems to be positive for both Kelvin probe force images in Figs.3b and 3e. Is this discrepancy due to a mistake on labeling the scale bar?

Reviewer #3 (Remarks to the Author):

This is an interesting and original paper. It shows high-resolution AFM and KPFM data on anthracene and a halogenated anthracene derivative, H substituted by F and Cl. The subject of investigation is the pi-hole expected in the halogenated molecule. The halogenated molecule is by theory expected to exhibit so-called pi-holes above the center of the carbon rings, that is, (I think)

regions of positive electrostatic potential perpendicular the molecular framework, while anthracene is not expected to show pi-holes.

From the comparison of the KPFM images, the authors argue to have demonstrated the first experimental proof and first visualization of a pi hole. In addition, the differences in adsorption height of the two molecules are measured, showing a larger adsorption height of the fluorinated molecule. I think some citations to previous works are missing. The paper and the subject of investigation are very interesting, the measurements are of very high quality, and I support publication. But I think it should be made clearer what can be deduced exactly from the experiment, and I recommend some additional citations and discussion in their context.

I think, several citations (however, all to works of my group) are missing. I like to bring those to the attention of the authors. They might have some impact on the paper.

N. Moll, B. Schuler, S. Kawai, F. Xu, L. Peng, A. Orita, J. Otera, A. Curioni, M. Neu, J. Repp, G. Meyer, L. Gross, *Nano Lett.* 14, 6127 (2014) shows a very similar contrast inversion by KPFM for such molecular motives (fluorinated against hydrogenated C6 rings, see Fig. 2 of that paper), as discussed in the paper under review and assigned to a pi-hole. I believe it should therefore be cited with this mentioned. The 2014 paper was not discussed in the context of pi-holes though. In Figure. 3: Larger area images in b) and e) would be nice. Then one might see, whether there is a red halo around the blue features in b) and a blue halo around the red features in e). cf. *Nano Lett.* 14, 6127, 2014. Does the background LCPD signal shift so much by going only 30 pm closer? Note that besides the molecule the LCPD (background?) in b) is about 150 mV, while in e) it is about 0 mV. This might also be related to the halo observed in *Nano Lett.* 14, 6127, 2014. I understand that in the monolayer that might lead to different results compared to isolated molecules. For the calculated images at least, larger frames can be shown, e.g. in the SI, to see if the halo is seen there or not.

In Fig. 3 c) f) why are no units given in the calculation? What is the background? Do plots c and f have the same color scale, with same offset? I suggest plotting the two calculated images with identical color scale, i.e., same color for same values.

Are these molecules in Fig. 3 (exp. and theory) in the self-assembled monolayer as in Fig S3? Or single isolated molecules? Should be mentioned.

A work that highlights some of the difficulties in interpreting and simulating KPFM images, in particular on molecules and that could be important in the context of this work: B. Schuler, S.-X. Liu, Y. Geng, S. Decurtins, G. Meyer, L. Gross, *Nano Lett.* 14, 3342 2014. This work indicates that it could be better to compare KPFM maps to the electrostatic field rather than the potential energy landscape (of the system without the presence of the tip). This is important in the context of the interpretation of the KPFM data and the claim of direct observation and proof of the pi-hole. For the interpretation it could be important if contrast reflects qualitatively the potential of the E-field. But I understand that the detailed discussion of what exactly is measured by KPFM is not the scope of the paper.

In the context of the adsorption-height determination by AFM, the introduction of that technique being used here should be cited, that is: B. Schuler, W. Liu, A. Tkatchenko, N. Moll, G. Meyer, A. Mistry, D. Fox, L. Gross, *Adsorption Geometry Determination of Single Molecules by Atomic Force Microscopy.* *Phys. Rev. Lett.* 111, 106103, 2013. There it is also shown, that the difference of $df_{\min}(z)$, that is of z^* of molecule and metal surface is not a very exact value for determining the absolute adsorption height. For a CO tip and a Cu(111) surface we found an offset $z^*_{\text{off}} = 0.8$ Angstrom that should be applied to the difference. However, it could be different with different tips and on Au(111). And also the relatively large oscillation amplitude of 200 pm could contribute. Are the measurements on the different positions in Fig. 4a started from the very same setpoint? There is no information on the location, nor the STM values for the setpoint provided. Consider that the difference of z^* on surface and molecule does not directly reflect the adsorption height. Importantly, it becomes not very clear to me, what can be concluded exactly from the KPFM measurements, in terms of the pi-hole. Does this prove the pi-hole? These abstract and conclusions are very bold about it. However, trying to find where that was proven in the main text it reads: "This internal variation of the LCPD signal also nicely matches the distribution of the electrostatic potential calculated for a free-standing FCI-An molecule, shown in Fig. 3a. This effect is associated with the heterogeneous charge distribution of the π -hole due to the different electronegativity of fluorine and chlorine modulating the charge transfer from the π -system locally." That reminded me of the discussion about the H bonds in the AFM community. If you know it is there, and you see contrast that relates to it and matches, does that mean you have imaged

it?

I think it is important to state the definition of a pi hole. The term is relatively new and I am not sure about the definition. I think its: "A pi hole is a region of positive electrostatic potential that is perpendicular the molecular framework" no? Or is a relative maximum in the electrostatic potential that is perpendicular the molecular framework enough (i.e., can it still be a negative value of electrostatic potential?). If it is the former definition, i.e., a region of positive (!) electrostatic potential, I wonder how that can be proven by KPFM, which is only a qualitative technique. That is, I can only compare values, but how do I find out wheather the potential is positive or negative (e.g. at the center of the fluorinated ring)? This point might be addressed in a revision.

The authors write: "In previous work, we demonstrate that the sensitivity of the KPFM method can be substantially enhanced^{9,16} using functionalized probes." In these two cited works they reused the same tip functionalization and methods (CO and Xe) that we used several years before for demonstrating that the sensitivity of the KPFM method can be substantially enhanced using functionalized probes. That is F. Mohn, L. Gross, N. Moll, G. Meyer; Nat. Nano. 7, 227, 2012. For Xe tips: F. Mohn, B. Schuler, L. Gross, G. Meyer, Appl. Phys. Lett. 102, 73109, 2013. Obviously, the authors are aware of our work which technique they applied, and they also cite some of it in a different context. But why do they write that they showed that the KPFM method can be substantially enhanced using functionalized probes?

I think the paper should be published after revisions, but I would ask the authors to take care with the argumentation and their conclusions, and improve the argumentation. Given the qualitative nature of KPFM, proving a positive value of the electrostatic potential might not be possible from the experiment alone. However, even if the experimental proof cannot be made directly, I think the paper should be published. I think more important than a bold claim is an honest discussion and providing clear arguments.

I know how that report looks like. I request many of my own works to be cited. It is not because I want more citations, it is because I think they are important in the context. I sign the report for clarity.

Signed report: Leo Gross

Reviewer #4 (Remarks to the Author):

This paper is of great importance. The authors show experimental evidence of the n-holes in 9,10-dichlorooctafluoroanthracene and their absence in anthracene. I have only minor comments for the authors:

1) Since the authors begin by comparing the electrostatic potentials of benzene and perfluorobenzene, they may be interested in a paper published in 2015 [J. Comput. Sci. 10, 209 (2015)] which looks are a range of benzene derivatives and their interations with both the positive and negative ends of HCN.

2) The concept of the n-hole has evolved since the paper cited as reference 3. The authors may be interested in a recent paper in PCCP, 23, 16458 (2021) and references cited therein.

RESPONSE TO REVIEWERS' COMMENTS

Referee comments are in italic font, our answers to them are in blue.

REVIEWER COMMENTS

Reviewer #1 (Remarks to the Author):

This article describes the experimental observation of the pi-hole, and pi-hole is indeed an important concept, whether in guiding noncovalent interactions or in solid surface adsorption, it has also shown its importance in the field of crystal engineering. The experiments in the article are completed, and the discussion on results is also sufficient.

However I noticed that the experimental results of the article (Fig3) actually correspond to the electron density. Therefore, due to the lack of direct correspondence between the change of electrostatic potential map of molecule and the change of electron density (J. Chem. Theory Comput. 2009, 5, 2301 – 2312), I suggest that all the molecular surface electrostatic potential maps be changed to "electron density difference maps", or to present the electronic density difference maps together with the molecular surface electrostatic potential map, which is not easy to cause confusion for readers.

Another issue is that when the hydrogen atom is replaced by the fluorine atom, it is not necessary to emphasize the decrease of pi-electron density, but the decrease of electron density along the central position perpendicular to hexafluorobenzene pi-plane.

Finally, the article in communication form can be accepted for publication after minor revision.

We thank the referee for the positive assessment of our work. Regarding the concern raised by the reviewer, we agree that the comparison based on electrostatic potential maps suffers from the problem of different electron densities of the molecules. However, we would like to stress that the electron difference maps also cannot provide a complete picture of π -hole as it takes into account not only charge transfer due to different electronegativity but also different number valence electrons of substituent species

Moreover, the experimental KPFM images (Fig. 3b,e) are compared to simulated KPFM images (Fig. 3c,f), where there is a one-to-one correspondence. A detailed description of the theory behind calculated KPFM images was described in a previous publication (B. Mallada et al. 2021, Science 374, 863–867).

Figure R1. Electron density difference map of 9,10-dichlorooctafluoroanthracene versus anthracene [$\Delta\rho = \rho(\text{C}_{14}\text{Cl}_2\text{F}_8) - \rho(\text{C}_{14}\text{H}_{10})$]. The blue color shows decreasing electron density, red color shows increasing electron density.

*ACTION: we added a short comment to the revised manuscript as well as new **Figure S2** according to the referee's suggestion.*

Reviewer #2 (Remarks to the Author):

The manuscript reports the direct visualization of pi-electron-deficient cavity (referred in the following as a pi-hole) in a 9,10-Dichlorooctafluoroanthracene (FCl-An) molecule deposited on a Au(111) surface using atomic force microscopy (AFM). This work seems to be the first direct experimental observation of the existence of a pi-hole, and therefore worth publishing in a high-profile journal. The observation is made by a carefully planned set of experiments combining scanning tunneling microscopy, and probe-functionalized non-contact atomic force microscopy operated in constant-height imaging and Kelvin probe force modes. The experimental results seem to be fairly supported by first principles calculations. To demonstrate that the observation of the pi-hole at the FCl-An molecules is not due to the interaction with the substrate, anthracene (An) molecules are co-adsorbed on the surface and the AFM signals are simultaneously compared over the self-assembled structure formed by the two molecules.

Both experimental and theoretical results are convincing and support, in my view, the conclusions raised by the authors. I therefore recommend publication of the manuscript in Nature Communications upon minor revisions.

Fig.S2 does not fully support that the mere deposition of An molecules does not result on the formation of self-organized molecular structures, as the coverage of An molecules in Fig.S2 seems

to be at least 50 times lower than the coverage of An molecules in Fig. 1. I wonder whether structures of An molecules could be formed when the molecules are deposited on a gold surface at room temperature at similar coverage to the one in Fig.1.

We appreciate the careful review of the manuscript and valuable comments of our work by the referee.

The deposition of An molecules at room temperature (RT) in a high coverage regime was not performed. However, we conducted a deposition of FCl-An molecules in both high and low-coverage regimes for the purpose of comparison. In the high coverage regime, FCl-An molecules exhibited a self-assembled two-dimensional (2D) island structure with a herringbone pattern (see Figure R2 top row below). In the low coverage regime at RT, FCl-An molecules primarily attached to the step edges, while no molecules were observed in the middle of the terraces due to significant thermally activated diffusion (not shown). Consequently, we conducted low-coverage deposition at a lower temperature, below RT. To cool the sample, it was extracted from the microscope at 4.5 K, following which it was exposed to the molecular gas for a few seconds. Although the precise temperature of the sample cannot be determined, given the rapid nature of the process, we estimate it to be approximately 100 - 200 K.

In contrast to the case of An molecules, which were observed as individual entities on the surface, FCl-An molecules formed small molecular clusters, indicating a high propensity for assembly (see **Figure R2** bottom row below). Additionally, single FCl-An molecules were exceptionally rare and consistently found attached to surface impurities.

Figure R2 Topographic and high-resolution AFM images of high (top) and low (down) coverage of FCl-An molecules on Au(111). Low coverage is achieved by decreasing the temperature of the sample during the deposition below RT

It is stated that “The anthracene molecule possesses a uniform positive LCPD signal, representing a negative charge distribution, while the fluorinated counterpart exhibits a bow-tie shape of negative LCPD, indicating a positive charge distribution”. However, the experimental LCPD value seems to be positive for both Kelvin probe force images in Figs.3b and 3e. Is this discrepancy due to a mistake on labeling the scale bar?

In the experimental LCPD, there is a total offset resulting from the macroscopic contact potential difference between the tip and the surface. On the other hand, the scale of the simulated LCPD is rather sensitive to the model of the tip, although there are hardly any changes of contrast in the simulated images apart from constant factor rescaling upon adjustment of the CO tip model polarizability. Consequently, while we cannot be sure about the magnitude of the charge redistribution in the molecules, we can still safely conclude where we see the respective locations of charge depletion and charge enhancement. We can check that the π -hole indeed corresponds to a positive charge by comparing the π -hole region to the expected σ -hole location on the Cl atoms: The observed LCPD value is roughly the same and corresponds to a more-positive-than-average charge density. In addition, please see also our reply to a similar comment of referee #3, where we justify the presence of the positive charge associated with π -hole.

Figure R3. LCPD maps on two FCl-An molecules for a far ($z=0$ pm) and close ($z=-20$ pm) tip-molecule distances. In both cases, the LCPD values on the Cl atoms, displaying a σ -hole, are similar to the measured LCPD of the π -hole within both FCl-An molecules.

Fig. 3 | Simulated and experimental charge distributions of FCl-An and An. (a, d), Electrostatic potential maps of free standing FCl-An and An molecules. **(b, e),** Experimental LCPD maps of FCl-An and An acquired with the same CO-tip in constant height mode. **(c, f),** Probe Particle simulated LCPD maps of FCl-An and An.

ACTION: We have updated the simulated KPFM maps in Figure 3 in the main text and we changed the section Methods accordingly.

Reviewer #3 (Remarks to the Author):

This is an interesting and original paper. It shows high-resolution AFM and KPFM data on anthracene and a halogenated anthracene derivative, H substituted by F and Cl. The subject of investigation is the pi-hole expected in the halogenated molecule. The halogenated molecule is by theory expected to exhibit so-called pi-holes above the center of the carbon rings, that is, (I think) regions of positive electrostatic potential perpendicular the molecular framework, while anthracene is not expected to show pi-holes.

From the comparison of the KPFM images, the authors argue to have demonstrated the first experimental proof and first visualization of a pi hole. In addition, the differences in adsorption height of the two molecules are measured, showing a larger adsorption height of the fluorinated molecule. I think some citations to previous works are missing. The paper and the subject of investigation are very interesting, the measurements are of very high quality, and I support publication. But I think it should be made clearer what can be deduced exactly from the experiment, and I recommend some additional citations and discussion in their context.

We appreciate the careful review of the manuscript and valuable comments of our work by the referee.

I think, several citations (however, all to works of my group) are missing. I like to bring those to the attention of the authors. They might have some impact on the paper. N. Moll, B. Schuler, S. Kawai, F. Xu, L. Peng, A. Orita, J. Otera, A. Curioni, M. Neu, J. Repp, G. Meyer, L. Gross, Nano Lett. 14, 6127 (2014) shows a very similar contrast inversion by KPFM for such molecular motives (fluorinated against hydrogenated C6 rings, see Fig. 2 of that paper), as discussed in the paper under review and assigned to a pi-hole. I believe it should therefore be cited with this mentioned. The 2014 paper was not discussed in the context of pi-holes though.

We thank the referee for this valuable comment.

ACTION: We added a comment mentioning the previous work in the context of π -hole.

In Figure. 3: Larger area images in b) and e) would be nice. Then one might see, whether there is a red halo around the blue features in b) and a blue halo around the red features in e). cf. Nano Lett. 14, 6127, 2014. Does the background LCPD signal shift so much by going only 30 pm closer? Note that besides the molecule the LCPD (background?) in b) is about 150 mV, while in e) it is about 0 mV. This might also be related to the halo observed in Nano Lett. 14, 6127, 2014. I understand that in the monolayer that might lead to different results compared to isolated molecules. For the calculated images at least, larger frames can be shown, e.g. in the SI, to see if the halo is seen there or not.

We refer the reviewer to **Figure S3** in SOM, which displays the requested large-scale KPFM images. In these large-scale KPFM images, we do not observe any halo effect due to close-packed molecular self-assembly.

Figure R4. (Left) LCPD map, acquired with a CO-tip, over an area covering six FCI-An molecules, displaying the characteristic bow-tie shape of the positive charge distribution of the π -hole and one central anthracene molecule exhibiting a featureless negative charge distribution. (Right) Δf^* map of the same region.

In Fig. 3 c) f) why are no units given in the calculation? What is the background? Do plots c and f have the same color scale, with same offset? I suggest plotting the two calculated images with identical color scale, i.e., same color for same values. Are these molecules in Fig. 3 (exp. and theory) in the self-assembled monolayer as in Fig S3? Or single isolated molecules? Should be mentioned.

We thank the referee for this comment. In the original version of the manuscript, we did not put the absolute scale of the simulated images because the overall scale depends beside the short-range linear component also on the long-range parabolic component of the KPFM signal, which cannot be evaluated from atomistic simulations. However, to facilitate the direct comparison between experimental and theoretical KPFM images, we decided to take fitted values for the long-range part described by the Kelvin parabola from experimental measurements at far tip-sample distances, where the short-range component is negligible.

For a detailed discussion of the short-range and long-range components of KPFM images as well as the theoretical model of KPFM simulation, we kindly refer the reviewer to a section in Supplementary material of *Mallada, B., Gallardo, A., Lamanec, M., de la Torre, B., Špirko, V., Hobza, P., & Jelinek, P. (2021) Science (Vol. 374, Issue 6569, pp. 863–867)*, where the detailed description of the model is provided.

Fig. 3 | Simulated and experimental charge distributions of FCl-An and An. (a, d), Electrostatic potential maps of free standing FCl-An and An molecules. (b, e), Experimental LCPD maps of FCl-An and An acquired with the same CO-tip in constant height mode. (c, f), Probe Particle simulated LCPD maps of FCl-An and An.

A work that highlights some of the difficulties in interpreting and simulating KPFM images, in

particular on molecules and that could be important in the context of this work: B. Schuler, S.-X. Liu, Y. Geng, S. Decurtins, G. Meyer, L. Gross, Nano Lett. 14, 3342 2014. This work indicates that it could be better to compare KPFM maps to the electrostatic field rather than the potential energy landscape (of the system without the presence of the tip). This is important in the context of the interpretation of the KPFM data and the claim of direct observation and proof of the pi-hole. For the interpretation it could be important if contrast reflects qualitatively the potential of the E-field. But I understand that the detailed discussion of what exactly is measured by KPFM is not the scope of the paper.

As discussed above, we developed an atomistic model, which describes the main mechanism of KPFM contrast on atomic scale. In principle, the atomic scale contrast emerging on short-range distances is given by a describing interaction between the static and induced charge on tip and sample. This term is linear in applied bias and causes displacement of the local contact potential difference according to the electrostatic interaction between static and induced charges of tip and sample (i.e. a charge interaction with a corresponding potential). Therefore, this makes a direct comparison of KPFM images to simple electrostatic potential or electric field of molecules difficult.

ACTION: None. We feel that this discussion is too technical and limited to a narrow group of experts. Thus, its presence would distract the reader from the main focus of the manuscript. We plan to prepare another manuscript where such a discussion of the origin of atomic scale contrast of KPFM will be discussed. Moreover, a detailed description of the model and imaging mechanism is described in the previous publication.

In the context of the adsorption-height determination by AFM, the introduction of that technique being used here should be cited, that is: B. Schuler, W. Liu, A. Tkatchenko, N. Moll, G. Meyer, A. Mistry, D. Fox, L. Gross, Adsorption Geometry Determination of Single Molecules by Atomic Force Microscopy. Phys. Rev. Lett. 111, 106103, 2013. There it is also shown, that the difference of $df_{min}(z)$, that is of z^ of molecule and metal surface is not a very exact value for determining the absolute adsorption height. For a CO tip and a Cu(111) surface we found an offset $z^*_{off} = 0.8$ Angstrom that should be applied to the difference. However, it could be different with different tips and on Au(111). And also the relatively large oscillation amplitude of 200 pm could contribute. Are the measurements on the different positions in Fig. 4a started from the very same setpoint? There is no information on the location, nor the STM values for the setpoint provided. Consider that the difference of z^* on surface and molecule does not directly reflect the adsorption height.*

We admit that there is a notable uncertainty in the experimental determination of absolute adsorption height, as described in the suggested citation of PRL111, 106103 (2013). Our main point was any way to show the relative difference in adsorption height between the two molecules. In this case, we measure the AFM frequency shift above the geometric center of

the molecule (above the interior of the middle hexagon of the anthracene, see Figure R5 below). The chemical composition of the sample in the vicinity of the tip is therefore similar, the only important difference being the electrostatic effects connected with the π -hole. As the CO tip carries a relatively small charge on the terminal O atom (based on our experience with the simulation of AFM images with CO-tip we consider a relatively small quadrupolar charge distribution on CO) and, moreover, the $df(z)$ curves were measured at a low bias (3 meV) so that the effect of tip and sample polarizability also remains small, we expect the impact of non-topographic effects on the measured height difference to be negligible. The observed agreement between this experimental height difference and the difference in the DFT-predicted adsorption heights is therefore not surprising. The expected correspondence between the AFM measurement and the true geometry concerning the height differences is also confirmed by using our Probe Particle simulator on the DFT-calculated geometry: The predicted vertical distances of frequency shift minimum z^* (for the tip oscillation amplitude of 200 pm) differ by ≈ 20 pm between the two molecules. Crucially, this prediction of the probe Particle model does not change if we swap our tip model that carries a d_z^2 -like charge quadrupole of $-0.1 e \text{ \AA}^2$ for a tip model without any electric charge. Such insensitivity validates the negligible role of electrostatics in this topography estimation.

Figure R5. Experimental and PP-simulated $\Delta f(z)$ spectroscopies over the central and outer rings of An (black and red solid lines) and the central and outer rings of FCl-An (purple and blue solid lines).

ACTION TO DO: We added the reference PRL111, 106103 (2013) as well as **Figure R5** to Supplementary materials.

Importantly, it becomes not very clear to me, what can be concluded exactly from the KPFM measurements, in terms of the π -hole. Does this prove the π -hole? These abstract and conclusions

are very bold about it. However, trying to find where that was proven in the main text it reads: "This internal variation of the LCPD signal also nicely matches the distribution of the electrostatic potential calculated for a free-standing FCl-An molecule, shown in Fig. 3a. This effect is associated with the heterogeneous charge distribution of the π -hole due to the different electronegativity of fluorine and chlorine modulating the charge transfer from the π -system locally." That reminded me of the discussion about the H bonds in the AFM community. If you know it is there, and you see contrast that relates to it and matches, does that mean you have imaged it?

I think it is important to state the definition of a pi hole. The term is relatively new and I am not sure about the definition. I think its: "A pi hole is a region of positive electrostatic potential that is perpendicular the molecular framework" no? Or is a relative maximum in the electrostatic potential that is perpendicular the molecular framework enough (i.e., can it still be a negative value of electrostatic potential?). If it is the former definition, i.e., a region of positive (!) electrostatic potential, I wonder how that can be proven by KPFM, which is only a qualitative technique. That is, I can only compare values, but how do I find out wheather the potential is positive or negative (e.g. at the center of the fluorinated ring)? This point might be addressed in a revision.

We agree with the referee that KPFM measurements are only quantitative and themselves cannot say anything about the sign of the charge on the substrate.

As discussed above, the range of experimental LCPD values is defined by a total offset resulting from the macroscopic (long-range) contribution to the contact potential difference between the tip and the surface. On the other hand, in principle, we can calibrate the LCPD values to a known charge. In the case of π -hole, we have two options. The first option is purely experimental. It consists of the simultaneous measurement of an object with a known charge. In this case, it is possible to take advantage of the presence of a positively charged σ -hole on Cl, which can be partially visualized when the tip is sufficiently close. **Figure R3** displays two LCPD maps taken in close tip-sample distances, where the emergence of low (blue) LCPD signal is presented not only in the central part of the molecule but also in the area where the σ -hole on chlorine atoms is presented. Importantly, both the central region of the molecule and σ -hole show very similar LCPD values ~ 110 mV. Thus, this direct comparison enables us to demonstrate the presence of the positively charged region associated with the π -hole.

The second option is the calibration using theoretical calculations. In particular, we calculate electrostatic potential maps (see Figure 3a,d), which clearly show the presence of a positive charge in the central part of the FCl-An molecule. Subsequently, a good agreement between the experimental and simulated images enables the prediction of the qualitative determination of the charge.

Figure R3. LCPD maps on two FCl-An molecules for far ($z=0$ pm) and close ($z=-20$ pm) tip-molecule distances. In both cases, the LCPD values on the Cl atoms, displaying a σ -hole, are similar to the measured LCPD of the π -hole within both FCl-An molecules.

ACTION: We added a paragraph discussing the qualitative analysis of the positive charge of π -hole, including a new **Figure S7** in Supplemental material.

The authors write: "In previous work, we demonstrate that the sensitivity of the KPFM method can be substantially enhanced^{9,16} using functionalized probes." In these two cited works they reused the same tip functionalization and methods (CO and Xe) that we used several years before for demonstrating that the sensitivity of the KPFM method can be substantially enhanced using functionalized probes. That is F. Mohn, L. Gross, N. Moll, G. Meyer; Nat. Nano. 7, 227, 2012. For Xe tips: F. Mohn, B. Schuler, L. Gross, G. Meyer, Appl. Phys. Lett. 102, 73109, 2013. Obviously, the authors are aware of our work which technique they applied, and they also cite some of it in a different context. But why do they write that they showed that the KPFM method can be substantially enhanced using functionalized probes?

We thank the referee for pointing out the missing references.

ACTION: we added two references in the text.

I think the paper should be published after revisions, but I would ask the authors to take care with the argumentation and their conclusions, and improve the argumentation. Given the qualitative nature of KPFM, proving a positive value of the electrostatic potential might not be possible from the experiment alone. However, even if the experimental proof cannot be made directly, I think the paper should be published. I think more important than a bold claim is an honest discussion and providing clear arguments.

We are thankful for the positive assessment of the work and we hope to convince the referee by the argumentation that we are able to resolve the π -hole by KPFM method.

Reviewer #4 (Remarks to the Author):

This paper is of great importance. The authors show experimental evidence of the π -holes in 9,10-dichlorooctafluoroanthracene and their absence in anthracene. I have only minor comments for the authors:

We highly appreciate the positive assessment of our work by the referee.

1) Since the authors begin by comparing the electrostatic potentials of benzene and perfluorobenzene, they may be interested in a paper published in 2015 [J. Comput. Sci. 10, 209 (2015)] which looks at a range of benzene derivatives and their interactions with both the positive and negative ends of HCN.

We thank the referee for drawing our attention to this reference.

ACTION: we added the reference to the manuscript.

2) The concept of the π -hole has evolved since the paper cited as reference 3. The authors may be interested in a recent paper in PCCP, 23, 16458 (2021) and references cited therein.

We thank the referee for drawing our attention to this reference. One of the issues the paper emphasizes is electron density depletion as a hallmark of the π -hole in contrast to the positive charge region stressed in our paper. We discuss this issue in our reply to referee #1; see also **Figure R1** in this reply.

ACTION: we added the reference to the manuscript.

REVIEWER COMMENTS

Reviewer #2 (Remarks to the Author):

I am satisfied with the revision made by the authors that properly addresses my previous concerns. I recommend publication of the revised version of the manuscript.

Reviewer #3 (Remarks to the Author):

The authors thoroughly revised the paper, and I would recommend publication in Nature communications after the following points have been addressed and considered.

1) I remain very skeptic about the statement that the total sign of the charge can be confirmed by the KPFM experiments. For example, see significant shifts of the LCPD as a function of distance for neutral(!) adatoms, and their different contrast w.r.t. the background (there NaCl) in L. Gross et al, Science. 324, 1428–1431 (2009), Fig. 4. These experiments show, that the LCPD shift as a function of distance, and also the lateral contrast (compared to background, there NaCl), might be misleading and not suitable for assigning the charge.

The authors might consider wording more carefully if a positive charge can be evidenced by the experiment. May be the claim could be softened from "confirming" the charge, to "visualizing/imaging" the pi-hole by experiment. (With the pi-hole being confirmed by theory). E.g. in: "This direct comparison enables us to confirm the presence of a positively charged region associated with n-hole."

2) In that context, it would be good to add KPFM data as a function of distance. Ideally, including reference data on the pristine surface. Similar to L. Gross et al, Science. 324, 1428–1431 (2009), Fig. 4.

3) In the new Fig. S7 please indicate in the KPFM images, the positions at which the LCPD values listed in the associated table had been obtained.

4) In Fig S5, please indicate tip-height offsets.

Signed report

Leo Gross

RESPONSE TO REVIEWERS' COMMENTS

Reviewer #3 (Remarks to the Author):

The authors thoroughly revised the paper, and I would recommend publication in Nature communications after the following points have been addressed and considered.

1) I remain very skeptical about the statement that the total sign of the charge can be confirmed by the KPFM experiments. For example, see significant shifts of the LCPD as a function of distance for neutral(!) adatoms, and their different contrast w.r.t. the background (there NaCl) in L. Gross et al, Science. 324, 1428–1431 (2009), Fig. 4. These experiments show, that the LCPD shift as a function of distance, and also the lateral contrast (compared to background, there NaCl), might be misleading and not suitable for assigning the charge.

The authors might consider wording more carefully if a positive charge can be evidenced by the experiment. May be the claim could be softened from “confirming” the charge, to “visualizing/imaging” the pi-hole by experiment. (With the pi-hole being confirmed by theory). E.g. in: “This direct comparison enables us to confirm the presence of a positively charged region associated with π -hole.”

We thank the referee for his valuable comments. We agree that, in general, the exact determination of the sign of the charge using KPFM is generally debatable, as demonstrated in the case of the Au adatom on NaCl presented by the referee. However, we are convinced that under certain circumstances, it is possible to determine the absolute distribution of surface charges, as demonstrated in references 11,19 of the manuscript.

In this particular case, despite the very good agreement between experiment and theory, the absolute assignment of charge distribution is still disputable due to the missing reference measurements of $V_{cpd}(z)$ on the metal surface. For this reason, we have rephrased some sentences as requested by the referee to avoid any misleading interpretation of the capability of the KPFM technique to assign absolute charges on surfaces.

Action: We modified the main text highlighted in yellow.

2) In that context, it would be good to add KPFM data as a function of distance. Ideally, including reference data on the pristine surface. Similar to L. Gross et al, Science. 324, 1428–1431 (2009), Fig. 4.

We have added the evolution of V_{cpd} as a function of distance for both molecules in the supplementary figure **Fig S7**. Unfortunately, we do not have reference measurements on a bare metal surface.

Fig. S7 | LCPD vs z on FCl-An and An. (Left) LCPD vs z over the left hexagon ring of FCl-An (blue squares), central hexagon (blue circles), and the left hexagon ring of An (red squares) and central hexagon (red circles). (Right) LCPD and Δf^* maps for FCl-An and An in the XZ plane (line over a molecule and perpendicular to the surface) obtained with a CO-tip showing the position of the LCPD(z).

3) In the new Fig. S7 please indicate in the KPFM images, the positions at which the LCPD values listed in the associated table had been obtained.

Action: We modified Fig. S7 accordingly (now Fig.S8).

4) In Fig S5, please indicate tip-height offsets.

Action: We added tip-height offset in Fig. S5.

REVIEWERS' COMMENTS

Reviewer #3 (Remarks to the Author):

The authors added the requested KPFM data as a function of height and introduced a paragraph acknowledging the difficulties in assigning the sign of local charges by KPFM.

The authors wrote in their report "For this reason, we have rephrased some sentences as requested by the referee to avoid any misleading interpretation of the capability of the KPFM technique to assign absolute charges on surfaces."

I think they have done so in a few cases but in most not.

The authors introduced in the revised version a paragraph in which they address the challenge of the determination of the sign of the charge and write that the assignment of a positive charge is tentative. That is:

"It should be noted that KPFM measurements are only quantitative. Therefore, it is difficult to assign the absolute sign of the charge on the substrate from KPFM measurements. In this case, we can only tentatively assign the presence of the positive charge associated with n-hole in the central part of the FCI-An molecule. First, Figure S7 shows the variation of LCPD value with tip-sample distance over different positions of FCI-An and An molecules. In the case of FCI-An molecule, we observe a gradual decrease of LCPD signal with distance, while in the case of An molecule, the signal increases. According to the microscopic model¹¹, the decrease of the LCPD signal corresponds to the presence of a positive charge on the surface, while the increase of the LCPD signal indicates the presence of a negative charge on the surface. However, absolute determination of the sign is impossible due to the absence of a reference measurement on a clean surface at the same height"

I agree with that paragraph.

As I understand, to prove the pi hole one needs to prove an absolute positive charge. Or not? Does the pi-hole imply a positive local charge? I asked that in the first round already and think I did not get a clear answer. I requested that this point should be made clear in the MS. Also that could be more clear.

When the assignment of the charge sign is tentative, how is then the experiment a clear proof of the pi-hole? The authors should make that clear how this is a proof, when the sign cannot be proven, or also consider softening the sentences where they mention experimental proof of the pi hole.

The following statements seem to not reflect the tentative character of the assignment of a positive charge, and thus the tentative character of the experimental proof of a pi-hole. I ask the authors once more to consider rewording those.

Abstract: "Here we present the resolution of the π -hole on a single molecule using the Kelvin probe force microscopy, which confirms the theoretical prediction of its existence." (Maybe better, e.g., "supports" or "strongly supports" than "confirms"?)

Main text: "However, a direct experimental observation that would clearly prove the existence of the n-hole has been lacking so far." (Given the tentative character of the charge assignment by experiment, the work under review might not be called a clear experimental prove either.)

Main text: "Here, we experimentally confirm the existence of the n-hole in 9,10-Dichlorooctafluoroanthracene C₁₄F₈Cl₂ (FCI-An) molecule and, at the same time, its absence in anthracene C₁₄H₁₀ (An)."

Conclusion: "Here, we presented the experimental evidence of the existence of π -hole in molecular systems by its real space imaging on a single molecule by means of Kelvin probe force microscopy"

Minor points:

1) I guess, in the beginning of the new paragraph, the sentence:

"It should be noted that KPFM measurements are only quantitative."

Should read:

"It should be noted that KPFM measurements are only qualitative."

Or

"It should be noted that KPFM measurements are not quantitative."

Or maybe better:

"It should be noted that KPFM measurements on the atomic scale are only qualitative."

(As on larger length scales KPFM measurements, measuring CPD, are quantitative).

2) As a side note that does not necessarily require action:

The authors also added another argument that was previously only in the rebuttal letter, that is: "Second, we take advantage of the presence of a positively charged σ -hole on Cl, see Figure 2d. The σ -hole can be partially visualized when the probe is sufficiently close to the substrate, see Figure S8. Importantly, both the central region of the molecule and σ -hole show very similar LCPD values ~ 110 mV. This comparison tentatively indicates the presence of a positively charged region associated with π -hole."

The π -hole in this paper and the σ -hole in Science 374, 863–867 (2021) are facing the tip (oriented perpendicular to the surface). Here the σ hole of the Cl is oriented in a different direction (in the surface plane), and thus not facing the tip. Should this σ -hole be observable by KPFM, when it is basically imaged from the side, where it is surrounded by negative charge? Is it a good reference for the measurement? With the "tentative" been added, I do not mind if the authors want to bring this argument forward. Just wanted to mention why it did not convince me. The authors might consider changing the sentences about the experimental proof of the π hole mentioned above, and I would suggest publication. I do not want to delay publication further and feel sorry for being pedantic about the language. However, I feel that it is important to be precise in this instance.

Signed

Leo Gross

RESPONSE TO REVIEWERS COMMENTS

Reviewer #3 (Remarks to the Author):

The authors added the requested KPFM data as a function of height and introduced a paragraph acknowledging the difficulties in assigning the sign of local charges by KPFM.

The authors wrote in their report “For this reason, we have rephrased some sentences as requested by the referee to avoid any misleading interpretation of the capability of the KPFM technique to assign absolute charges on surfaces.”

I think they have done so in a few cases but in most not.

The authors introduced in the revised version a paragraph in which they address the challenge of the determination of the sign of the charge and write that the assignment of a positive charge is tentative. That is:

“It should be noted that KPFM measurements are only quantitative. Therefore, it is difficult to assign the absolute sign of the charge on the substrate from KPFM measurements. In this case, we can only tentatively assign the presence of the positive charge associated with π -hole in the central part of the FCI-An molecule. First, Figure S7 shows the variation of LCPD value with tip-sample distance over different positions of FCI-An and An molecules. In the case of FCI-An molecule, we observe a gradual decrease of LCPD signal with distance, while in the case of An molecule, the signal increases. According to the microscopic model¹¹, the decrease of the LCPD signal corresponds to the presence of a positive charge on the surface, while the increase of the LCPD signal indicates the presence of a negative charge on the surface. However, absolute determination of the sign is impossible due to the absence of a reference measurement on a clean surface at the same height”

I agree with that paragraph.

As I understand, to prove the pi hole one needs to prove an absolute positive charge. Or not? Does the pi-hole imply a positive local charge? I asked that in the first round already and think I did not get a clear answer. I requested that this point should be made clear in the MS. Also that could be more clear.

When the assignment of the charge sign is tentative, how is then the experiment a clear proof of the pi-hole? The authors should make that clear how this is a proof, when the sign cannot be proven, or also consider softening the sentences where they mention experimental proof of the pi hole.

The following statements seem to not reflect the tentative character of the assignment of a positive charge, and thus the tentative character of the experimental proof of a pi-hole. I ask the authors once more to consider rewording those.

Abstract: “Here we present the resolution of the π -hole on a single molecule using the Kelvin probe force microscopy, which confirms the theoretical prediction of its existence.” (Maybe better, e.g., “supports” or “strongly supports” than “confirms”?)

We rephrased it as follows:

Here we present the resolution of the π -hole on a single molecule using the Kelvin probe force microscopy, which supports the theoretical prediction of its existence.

Main text: “However, a direct experimental observation that would clearly prove the existence of the π -hole has been lacking so far.” (Given the tentative character of the charge assignment by experiment, the work under review might not be called a clear experimental prove either.)

Actually, this sentence refers to the past, so we do not feel that this is the wrong statement.

Main text: “Here, we experimentally confirm the existence of the π -hole in 9,10-Dichlorooctafluoroanthracene C₁₄F₈Cl₂ (FCI-An) molecule and, at the same time, its absence in anthracene C₁₄H₁₀ (An).”

We rephrased it as follows:

Here, we experimentally visualize the π -hole in 9,10-Dichlorooctafluoroanthracene C₁₄F₈Cl₂ (FCI-An) molecule and, at the same time, its absence in anthracene C₁₄H₁₀ (An).

Conclusion: “Here, we presented the experimental evidence of the existence of π -hole in molecular systems by its real space imaging on a single molecule by means of Kelvin probe force microscopy”

We rephrased it as follows:

Here, we presented the experimental imaging of π -hole in molecular systems by its real space imaging on a single molecule by means of Kelvin probe force microscopy.

Minor points:

1) I guess, in the beginning of the new paragraph, the sentence:

“It should be noted that KPFM measurements are only quantitative.”

Should read:

“It should be noted that KPFM measurements are only qualitative.”

Or

“It should be noted that KPFM measurements are not quantitative.”

Or maybe better:

“It should be noted that KPFM measurements on the atomic scale are only qualitative.”

(As on larger length scales KPFM measurements, measuring CPD, are quantitative).

We rephrased it as follows:

It should be noted that KPFM measurements on the atomic scale are only qualitative

2) As a side note that does not necessarily require action:

The authors also added another argument that was previously only in the rebuttal letter, that is:

“Second, we take advantage of the presence of a positively charged σ -hole on Cl, see Figure 2d. The σ -hole can be partially visualized when the probe is sufficiently close to the substrate, see Figure S8. Importantly, both the central region of the molecule and σ -hole show very similar LCPD values ~ 110 mV. This comparison tentatively indicates the presence of a positively charged region associated with π -hole.”

The π -hole in this paper and the σ -hole in Science 374, 863–867 (2021) are facing the tip (oriented perpendicular to the surface). Here the σ -hole of the Cl is oriented in a different direction (in the surface plane), and thus not facing the tip. Should this σ -hole be observable by KPFM, when it is basically imaged from the

side, where it is surrounded by negative charge? Is it a good reference for the measurement? With the “tentative” been added, I do not mind if the authors want to bring this argument forward. Just wanted to mention why it did not convince me.

Honestly, we disagree with the referee on this point. Actually, the presence of sigma hole is also visible on the electrostatic potential map, see Figure 2d, even though it has an in-plane orientation.

The presence of a positive charge of sigma-hole is also revealed in simulated KPFM images calculated in close tip-sample distance. Thus, we think this is a relevant argument, and we decided to keep it.